# Investigation of the Relationship between the S1 Domain and Its Molecular Functions Derived from Studies of the Tertiary Structure

**DOI:** 10.3390/molecules24203681

**Published:** 2019-10-13

**Authors:** Evgenia I. Deryusheva, Andrey V. Machulin, Maxim A. Matyunin, Oxana V. Galzitskaya

**Affiliations:** 1Institute for Biological Instrumentation, Federal Research Center “Pushchino Scientific Center for Biological Research of the Russian Academy of Sciences”, 142290 Pushchino, Moscow Region, Russia; evgenia.deryusheva@gmail.com; 2Skryabin Institute of Biochemistry and Physiology of Microorganisms, Russian Academy of Sciences, Federal Research Center “Pushchino Scientific Center for Biological Research of the Russian Academy of Sciences”, 142290 Pushchino, Moscow Region, Russia; and.machul@gmail.com; 3Institute of Protein Research, Russian Academy of Sciences, 142290 Pushchino, Moscow Region, Russia; krabovm@gmail.com; 4Institute of Theoretical and Experimental Biophysics, Russian Academy of Sciences, 142290 Pushchino, Moscow Region, Russia

**Keywords:** S1 domain, RNA-binding, structural repeats, 30 S ribosomal protein S1

## Abstract

S1 domain, a structural variant of one of the “oldest” OB-folds (oligonucleotide/oligosaccharide-binding fold), is widespread in various proteins in three domains of life: Bacteria, Eukaryotes, and Archaea. In this study, it was shown that S1 domains of bacterial, eukaryotic, and archaeal proteins have a low percentage of identity, which indicates the uniqueness of the scaffold and is associated with protein functions. Assessment of the predisposition of tertiary flexibility of S1 domains using computational and statistical tools showed similar structural features and revealed functional flexible regions that are potentially involved in the interaction of natural binding partners. In addition, we analyzed the relative number and distribution of S1 domains in all domains of life and established specific features based on sequences and structures associated with molecular functions. The results correlate with the presence of repeats of the S1 domain in proteins containing the S1 domain in the range from one (bacterial and archaeal) to 15 (eukaryotic) and, apparently, are associated with the need for individual proteins to increase the affinity and specificity of protein binding to ligands.

## 1. Introduction

As is known, many natural proteins fold amino acid chains into compact units, defining structural domains [1]. These domains usually correlate with biological activities, and many protein functions can be described as composed by separate domain structures [2]. For globular proteins, this fact facilitates the description, evolution and construction of single amino acid chains that make up a set of integrated biological functions.

S1 domain is one of the structural versions of the OB-fold, which is considered one of the “oldest” protein folds that are mutation resistant and able to adapt to the binding of a large number of ligands. S1 domain is a β-barrel with an additional α-helix between the third and fourth β-strands; the main function of this domain is RNA-binding [3]. A short helix in the S1 domain adjoins to the binding cleft in the protein structures and determines a strong preference for ssRNA, which distinguishes S1 domains from other OB-fold proteins [4,5] (Figure 1).

The S1 domain is widespread in various proteins in three domains of life. Recently, we showed that the S1 domain is mainly identified in bacterial proteins (on average about 83%), then 14% in eukaryotic proteins, and 2.8% in archaeal proteins [6]. One of the unique features of the S1 domain is its large number of repeating variations in proteins containing the S1domain. In general, proteins containing the S1 domain contain one copy of the S1 domain (75% of all proteins containing the S1 domain). However, some of the eukaryotic proteins containing the S1 domain contain 7-15 repeats. Variability of the number of repeats in eukaryotic proteins is apparently associated with the need for individual proteins and proteins in protein complexes to increase the affinity and specificity of protein binding to ligands [6].

In this study, we examined the features and flexibility of the predisposition of individual regions in the tertiary structures of typical S1 domains from bacterial, eukaryotic, and archaeal proteins (24 proteins), representing all the major protein families, to investigate the relationship between the S1 structure and its molecular functions.

## 2. Results and Discussion

### 2.1. Distribution of the S1 Domain between Organism Super-Kingdoms

Data on the number of the S1 domain and its taxonomy distribution between the three main domains of living organisms can be found in the RCSB PDB (https://www.rcsb.org) and PDBe bases (https://www.ebi.ac.uk/pdbe/). The total number of defined 3D protein structures containing the S1 domain is 216 (RCSB PDB) and 225 (PDBe) (22.09.19; the search was carried out according to S1 domain profiles in the Pfam database (PF00575), Section 3.1). The percentage of proteins containing the S1 domain in each of the domains of the living world is approximately the same for two databases. Basically, the S1 domain is identified in eukaryotic proteins (69%), then 22% in bacterial proteins, and 8% in archaeal proteins (PDBe database). Such a distribution is undoubtedly associated with many specific structures of exosomes and RNA polymerases from eukaryotes, which in many cases are represented by repeating structures with different spatial resolutions. However, as will be shown below, the exact functions of the S1 domain structure in these complexes in most cases have not yet been determined. In addition, according to the bases, the S1 domain is allocated in the K3L protein from the vaccinia virus (Wisconsin strain) [7].

In this study, we selected 24 proteins containing S1 domains from the three main domains of living organisms: Bacteria, Eukaryotes, and Archaea (in accordance with the Materials and Methods section) to further examine the relationship between the S1 domain and its functions originated from the tertiary structure. Our data set consists of S1 domains, which represent all the major protein families. Selected proteins were obtained by X-ray crystallography, electron microscopy with different resolutions, and NMR methods.

### 2.2. S1 domain in Bacterial Proteins

The most common bacterial proteins containing the S1 domain are polynucleotide phosphorylase (PNPase), ribonuclease (RNase R, RNase E, RNase G), transcription factor NusA, transcription accessory protein Tex, general stress protein 13 (GSP13), and the S1 ribosomal protein S1 family [6]. PNPase is a conserved, widely distributed phosphorolytic 3′-5′exoribonuclease, which may also function under certain circumstances as a template-independent RNA polymerase [8]. Using the S1 domain as an example of PNPase, it was shown that conserved residues F19, F22, H34, D64, and R68 are located on the surface of the domain and form a specific RNA-binding site [3]. Ribonucleases are the main exoribonuclease that interferes with all these fundamental processes [9]; ribonucleases can act independently or as a component of an exosome, an essential RNA-degrading multiprotein complex [10]. RNaseE and PNPase are part of a bacterial complex degradosomy involved in rRNA processing and mRNA degradation [11]. RNase R is an important participant in the process of mRNA degradation [12]. Endoribonuclease RNase G is responsible for the formation of the mature 5’ end of 16S rRNA [13]. Transcription factor NusA can increase the pause in termination and enhances factor-independent transcription termination in vitro. In addition, NusA modulates rho-dependent termination [14]. Tex (toxin expression) protein is an important protein involved in the expression of critical toxin genes [15]. In the Tex S1 domain, a mutation experiment showed that F668, F671, H683, and R718 are involved in RNA binding [16]. The level of GSP13 protein rises after heat shock, salt stress, ethanol stress, glucose starvation, oxidative stress, and starvation with ammonia [17,18]. In addition, the mRNA level and the level of GSP13 protein show a significant increase after cold shock [19]. Moreover, it was reported that GSP13 protein is associated with the ribosome only during the exponential growth phase, while its function in the ribosome remains unclear [20]. Alignment of the structure shows that the corresponding residues in GSP13, Y21, F24, H34, D66, and K71 are conserved and should be involved in the RNA binding [21]. The functions of the S1 domain in the considered bacterial proteins are given in Table 1.

In general, for all S1 domains in bacterial proteins, participation in RNA binding processes of was confirmed experimentally.

As demonstrated in our recent paper [6], the family of multifunctional 30S ribosomal proteins S1 makes up about 20% (available in the UniProt database sequences) of all bacterial proteins containing the S1 domain. A distinctive feature of this family is the presence of several repetitions of the S1 domain: from one to six. In addition, more detailed studies of the family of 30S ribosomal protein S1 showed that there is a correlation between the number of S1 domains in bacteria and their belonging to a certain phylum [22,23]. A biochemical experimental study of various fragments of the 30S ribosomal protein S1 from *Escherichia coli* made it possible to establish the functions of individual protein domains and parts. The correlation between the functions of individual domains and protein structures for this protein family will be discussed in more detail below (Section 2.7).

### 2.3. S1 domain in Eukaryotic Proteins

For eukaryotes, the S1 domain is identified in proteins belonging to the intracellular protein complex PM/Scl-complex (exosome), which is involved in RNA processing. The catalyst component in the exosome of the eukaryotic protein Rrp44 (Dis3p) binds RNA and orients it in the orthogonal direction to the exosome core for processing long RNA molecules [24]. These proteins increase the efficiency of RNA binding and RNA degradation and promote the interaction of exosome with RNA containing less adenine [25]. The eukaryotic protein Rrp5p is involved in the processing of pre-rRNA and maturation of the 40S and 60S ribosomal subunits [26]. Eukaryotic initiation factor 2 (eIF2) is a multifunctional protein with three subunits called eIF2α, eIF2β, and eIF2γ. The eIF2α subunit is the main site for controlling the translation initiation process by phosphorylation of a specific serine residue [27]. It contains the S1 motif domain, which is a potential, but not entirely defined, RNA binding function. The Pol II elongation complex in vitro requires the kinase function of the elongation factors PAF1 complex (PAF) and SPT6 protein. The exiting RNA in the activated Pol II–DSIF (sensitivity-inducing factor)–PAF–SPT6 elongation complex, passes through a positively charged groove formed between the S1 and RuvC-like domains of the SPT6 core [28].

Defined functions of the S1 domain in the considered eukaryotic proteins are given in Table 2.

The participation of S1 domains in processes associated with RNA interaction has been experimentally confirmed for some of the considered proteins. These interactions were established at the level of cellular systems in which these proteins are included.

### 2.4. S1 domain in Archaeal Proteins

Non-catalytic components of the archaeal exosome Rrp4 and Csl4 form the upper part of the central channel of the circular exosome structure. Exosomes become the central mechanisms of 3`-5` processing and degradation of RNA in archaea. The trimer of Csl4 or Rrp4 subunits forms the surface of multi-domain macromolecular interaction on the RNase-PH domain ring with central S1 domains and peripheral domains of KH and zinc-ribbon [29]. The heterotrimeric archaeal translation initiation factor (aIF) and the eukaryotic factor eIF2 homologous to aIF also contain the S1 domain in their structures. These proteins play a key role in the initiation of protein synthesis by transferring the initiator methionyl tRNA to the P site of the small ribosomal subunit [30,31].

Defined functions of the S1 domain in the considered archaeal proteins are given in Table 3.

Note that in bacterial proteins, the S1 domain is the main in the process of interaction with RNA; in higher organisms (eukaryotes and archaea), it almost always, in cooperation with other domains, forms a central channel for RNA movement. Interestingly, the domains involved in the formation of the RNA entry pore are homologous to the S1 domain, for example, the KH- [29], PH- [32], YqgF- [33,34] or CSD [35] domains, the main function of which is also to bind RNA.

### 2.5. Analysis of Consensus Sequence of S1 Domains from Bacteria, Eukaryotes, and Archaea

Analysis of consensus sequence of S1 domains from Bacteria, Eukaryotes, and Archaea will provide information on the position of critical amino acids and their structural and functional disturbance in various proteins containing S1 domains. As mentioned above (Section Introduction), the S1 domain is a small structural domain of 70 amino acid residues that folds into a five-stranded antiparallel β-barrel bounded by a single α-helix [3].

In one example of the S1 domain, it was shown that the conserved residues F19, F22, H34, D64, and R68 (positions are given for PNPase) are located on the surface of the domain and form a specific RNA-binding site [3,16,21]. The analysis of these amino acids from our investigated data set is useful for understanding the relationship between the structure, function, and evolution of proteins containing the S1 domain (Figure 2).

F23, F34, H36, D64, and R67 (positions labeled according to alignment profiles for S1 domains from Bacteria) are responsible for oligonucleotide binding and are highly conserved during evolution among bacterial proteins (Figure 2a). The forces responsible for this interaction are based on stacking and include the side chains of these residues like CSD domains [36]. The loop between the third and fourth (if there is no α-helix) β-strands for bacteria usually consists of 10–14 amino acids, in which the non-polar amino acids, G33 and L44, are highly conserved and must fulfill important functions (Figure 2a).

For S1 domains from archaea and eukaryotes, only three conserved residues were revealed (of the five residues mentioned): F28, D66 and R71 for archaea, F25, D68 and R71 for eukaryotes (Figure 2b,c). This fact may be associated with the identification of the S1 domain in archaeal and eukaryotic proteins that make up the intracellular protein complex (Section 2.3 and Section 2.4.). In addition, V and G residues of the β-strands of the S1 domain among bacterial, eukaryotes, and archaeal proteins are highly conserved and are considered critical in terms of functional activity (Figure 2a–c).

### 2.6. Identity of S1 Domains in the Bacterial, Eukaryotic, and Archaeal Proteins

Alignments of sequences of S1 domains allowed us to calculate the average identity of S1 domains in each domain of life and between them (Appendix A). As you can see, the S1 domains of bacterial, eukaryotic, and archaeal proteins have a fairly low percentage of identity. Among bacterial proteins, the largest percentage (51%) are of the S1 domain of general stress protein 13 (*B.subtilis*, PDB code: 2k4k) and the S1 domain of PNAse of *E.coli* (PDB code: 1sro) (exclude PNAse from *C. vibrioides* and *E.coli*). This percentage of identity is the highest value among all S1 domains alignments. For eukaryotic proteins, the highest identity (46%) belongs to the S1 domains from the subunit of RNA polymerase II from *K. phaffii* (PDB code: 6ir9) and the subunit RPB7 of DNA-directed RNA polymerase II from *S. cerevisiae* (PDB code: 4a3g) (exclude RNA polymerase II subunit G from *H. sapiens* and *K. phaffii*). Among archaeal proteins, the highest percentage of identity (43%) belongs to the S1 domain from the component of the exosome complex Rrp4 from (*A. pernix,* PDB code: 2z0s) and the S1 domain from the component of the exosome complex Csl4 (*S. solfataricus,* PDB code: 3l7z). The minimum percentage of identity is 5%, which is found between the component of the eukaryotic complex RRP40 from *H. sapiens* (PDB code: 2nn6) and exosome complex exonuclease RRP44 from *H. sapiens* (PDB code: 6h25).

3D model alignments between protein structures can suggests not only evolutionary and functional relationships, but also allows one to reveal differences that may not be distinguishable when comparing sequences [37]. The structural alignment of S1 domains from each domains of life and between them allowed us to compare their tertiary structure similarity (Appendix A). As can be seen, the tertiary structures of the S1 domains are very well aligned (Figure 3). The average root-mean-square deviation (RMSD) between the S1 domains in each domains of life and between them is 2Å, which is usually considered very close [38]. This result confirms the earlier assumption that for the general functioning of proteins with the S1 domain, the structure scaffold is more important than the amino acid sequence [6,22]. As can be seen from Figure 3, the main differences in the S1 domains in the considered proteins are in the loop’s regions of the domains. The flexibility of the loop regions and secondary structure regions will be discussed below (Section 2.7).

As mentioned above, the S1 domain is one of the “oldest” protein domains. Thus, the evolutionary analysis of S1 domains offers an example of how, through evolution, the ancient protein domain diverged in many different organisms and potentially plays a diverse role, primarily depending on the conserved mode of functioning. In addition, the occurrence of S1 domains in eukaryotes, archaea, and bacteria suggests their presence before the divergence between the three domains of life. For our data set, phylogenetic trees were constructed using the POSA [39] (Appendix A) and SALIGN [40] (Appendix A) servers based on the S1 domains sequences.

An analysis of the dendrograms obtained by the POSA and SALIGN servers (Appendix A) reveals that in some cases S1 domains from a eukaryotic protein (for example, 1wi5, 6h25, 2nn6) are obviously close to the bacterial S1 domains (5lm9, 5f6c, 4aim). Also, one archaeal S1 domain is evolutionary grouped with the eukaryotic S1 domain (1q46 and 1yz6). Others groups are associated with proteins belong to different or related protein families from bacteria (3bzc, 2k4k, 1sro), archaea (2z0s, 3l7z, component of Exosome complex), and eukaryotes (4a3g, 6ir9, polymerase family). These data show that from the very beginning of the evolution of a protein, a long polypeptide chain with a simple fold could disperse into various families that are not related to the sequence, but retain a unique fold with the same function.

### 2.7. Analysis of Structural Flexibility and Disorder of the S1 Domains

To analyze the structural flexibility of S1 domains, we used the FlexPred program, designed to predict the absolute per-residue fluctuations in a query protein from its 3D structure [41]. FlexPred uses the support vector regression (SVR) approach to predict the fluctuations of the Cα atom of each residue in the crystal structure of the query protein by considering the contact numbers of Cα atoms with a set of cutoffs and taking into account crystallographic B-factors [41]. FlexPred structural flexibility profiles and B-factors profiles for S1 domains are given in Appendix A.

The results of the analysis of some S1 structures for representative members of bacterial, eukaryotic, and archaeal proteins are shown in Figure 4. Analysis of the sequences of S1 domains from archaea, bacteria, and eukaryotes revealed that the loop regions, as expected, have more flexible residues. Loop region (in the region 40–50 a.a.) is systematically predicted to be more flexible in the S1 domains, and has the greatest tendency to intrinsic disorder.

According to the FlexPred program, the S1 domain from the exosome complex component Rrp4 is the most flexible S1 domain in archaea, 25% (PDB code: 2z0s; FlexPred). For Bacteria, the highest flexible S1 domain is identified in polynucleotide phosphorylase, 60% (PDB code: 4aim, FlexPred). For eukaryotic proteins, the most flexible S1 domain according to the FlexPred program is the S1 domain of the G subunit of RNA polymerase II (92%; PDB code: 6gmh).

Using the B-factor gives other results. So, the most flexible S1 domain (56%, B-factor) is indicated in the exosome complex exonuclease DIS3 from eukaryotes, PDB code: 2wp8. For archaeal proteins, the most flexible S1 domain was found in the exosome complex component Csl4 (52%, PDB code: 3l7z). For bacterial proteins, the highest flexible S1 domain is identified in the antitermination protein NusA (57%, PDB code: 5lm9, B-factor).

However, the use of the Kruskal-Wallis test for the data from Appendix A shows that there are no significant statistical differences between the percentage flexibility of the S1 domain in bacteria, archaea, and eukaryotes (*p*-value is 0.49798 for the B-factor; 0.49073 for FlexPred). So, in general, according to the profiles (Appendix A), S1 domains from bacterial, eukaryotic, and archaeal proteins are characterized by average values of flexibility (about 27%).

A comparison of the radius of gyration for S1 domains as an indicator of the compactness of the domain structure [42] of bacterial, eukaryotic, and archaeal proteins (Appendix A) also does not indicate significant statistical differences (*p*-value is 0.3961) with the average value of 12Å.

Amino acids can be classified as ‘order-promoting’ (C, W, F, I, Y, V, L, H, T, and N) and ‘disorder-promoting’ (D, M, A, R, G, Q, S, P, E, and K) residues according to their abundance in ordered proteins and disordered proteins [43,44]. Analysis of the content of amino acid residues in the S1 domains for our data set (Appendix A) revealed that the content of the ‘disorder-promoting’ residues is in the range from 51% to 61% and is stable for the S1 domains for archaeal, bacterial, and eukaryotic proteins.

As we have recently shown, the S1 domain repeats presence in proteins containing the S1-domain (according to the analysis of amino acids of proteins with S1 domains): in the range from one to six for bacterial proteins, almost always in one copy in archaeal proteins, and up to 15 for eukaryotic proteins [6]. Apparently, the number of such structural repeats of S1 and the degree of its rate of the flexibility are associated with the need for individual proteins to increase the affinity and specificity of protein binding to ligands. Thus, the number of possible functions increases due to an increase in the number of structural domains, and not a change in the characteristics of an individual structural unit, S1 domain. The features of certain proteins containing several S1 domains will be discussed below (Section 2.8).

### 2.8. Different Number of Structural S1 Repeats in Proteins and Its Molecular Functions

As we mentioned above, all archaeal proteins contain one copy of the S1 domain (excluding the family of N-terminal domains of the DNA replication initiator (cdc21/cdc54), PROSITE ID: PS50051), while the number of repeats in the eukaryotic proteins varies from 1 to 15. In bacterial proteins, the number of repeats is not more than 6, regardless of the protein length [6]. Currently, there are only a few defined structures of proteins containing more than one S1 domain.

First, the bacterial structure of Qβ replicase, which is responsible for the replication and transcription of Qβ viral RNA, consists of virus-encoded RNA-dependent RNA polymerase (β-subunit), EF-Tu, EF-Ts and the N-terminal part of the ribosomal protein S1. In S1, which is capable of initiating Qβ RNA replication, two N-terminal S1 domains are defined (PDB code: 4q7j) [45,46]. Structural and biochemical studies have shown that these domains fix the S1 protein on the β-subunit. The N-terminal region containing the first two S1 domains stably interacts with the core of Qβ replicase. For this structure, the first domain is predicted to be more flexible (Appendix A) than the second, which corresponds to mentioned structure (Figure 5a). In addition, these domains are linked by a flexible linker, providing additional structural mobility. In addition, each of these domains contained loop regions (40–50 a.a.), which are characterized by a high degree of mobility (Appendix A). Thus, these domains are necessary for the full-sized S1 protein to interact with the core of Qβ replicase and cooperatively mediate this interaction [45] (Figure 5a).

Recently, 70S ribosome was visualized by ensemble cryo-electron microscopy [47]. The cryo-electron microscopy was also used to obtain the structure of the inactive conformation of the S1 protein (four domains) as part of the hibernating 100S ribosome [48] (Figure 5b). As can be seen, only four structural domains (1,2,4,5) were determined (Figure 5b).

The multi-functional ribosomal protein S1 is part of the 30S ribosomal subunit. Proteins of the 30S ribosomal protein family interact with mRNAs, participate in the initiation and translation of mRNAs in vivo, and interact with the mRNA-like part of the tmRNA molecule. Similar to some other ribosomal proteins, ribosomal protein S1 is an autogenous repressor of its own synthesis. From the study of the structure of S1 on the 70S ribosome, it was shown that S1 interacts with other ribosomal proteins (S2, S3, S6, and S18) forming a dynamic network near the output and input channels of the mRNA in order to modulate the binding, folding, and movement of mRNA. In addition, the Protein Data Bank contains 3D structures of separate domains of ribosomal S1 from *E.coli* obtained by NMR [23,49].

As can be seen from Appendix A, the S1 domains for this structure are characterized by a similar percent of structural flexibility (63 and 50% for the first and second domains, FlexPred, respectively). The FlexPred profiles are analogous to the first, second (Appendix A), and fourth domains. Among them, the first domain can be considered as the most flexible (Appendix A). The profile for the firth domain (the most stable according to FlexPred) can be explained by the low resolution of the studied structure and the secondary structures have not been determined. Experimental studies of this protein showed that cutting one S1 domain from the C-terminus (i.e., the first domain) or two S1 domains from the N-terminus (the firth and the sixth) of a protein reduces only the efficiency of the protein functions, but not its function capabilities. In addition, cutting off the fifth and sixth domain leads to the effective participation of the remaining part of the protein only in the translation of synthetic mRNA [4,50]. These facts allowed us to assume a high degree of flexibility for the sixth domain (similar to the fifth domain) and confirm the need for a more stable core domain (second, third, and fourth) for the implementation of the basic functions of the protein. Our recent studies of all available S1 sequences (the UniProt database) showed that individual ribosomal S1 domains and full-sized proteins have the same organization [22,51]. The Percent of intrinsic flexibility was less for the central domains in multi-domain proteins. These facts suggest that for all multi-domain S1 proteins, a more stable and compact domain is located in the central part and is vital for RNA interaction, and more flexible terminals domains for other functions [51]. One of the reasons for the lack of 3D structures for the third S1 domain (more stable according to our data from [22]) may be the high mobility of this domain relative to other structural domains (Figure 5b).

In addition, we have shown that the unique S1 protein family differs in the classical sense from a protein with tandem repeats, such as the ANK family, leucine-rich-repeat proteins, etc. [52], which usually have an elongated shape. The family of ribosomal proteins S1 is apparently closed for organizing “beads on a string”, with each repeat folding into a globular domain, for example, Zn-finger domains [53], Ig-domains [54], and human matrix metalloproteinase [55]. So, these facts indicate a unique structural organization of the proteins of this family. The organization is closer to the formation of the quaternary structure of globular proteins, with the same structural organization of individual structural domains.

## 3. Materials and Methods

### 3.1. Construction Dataset of the Protein Containing S1 Domains

3D structures of proteins containing S1 domains were selected from the PDBe (https://www.ebi.ac.uk/pdbe/), European resource for the collection, organization, and dissemination of 3D structural data (from PDB and EMDB) on biological macromolecules and their complexes according to S1 domain profiles in the Pfam database (PF00575) and PROSITE (PS50126).

### 3.2. Analysis of S1 Domain Structures

A global pairwise sequence alignment (Needleman-Wunsch algorithm) using a dynamic programming algorithm was used. The Multiple Sequence Alignment was implemented by the Clustal Omega service (https://www.ebi.ac.uk/Tools/msa/clustalo/). Per-residue structural flexibility profiles for chosen S1 domain were obtained using the FlexPred software available at http://kiharalab.org/flexPred/ [41]. Structural alignment was done with the interactive multiple protein structure alignment server POSA (http://posa.godziklab.org), using multiple flexible structure alignment with partial order graphs [39]. This program allows to align maximum 20 structures at once (Appendix A contain 18 proteins, exclude S1-domains from the same proteins). Search, collection, alignment, representation and analysis by the described methods of the data were realized using the freely available programming language Python 3.6 (https://www.python.org/) as implemented in the PyCharm community edition 2017 (https://www.jetbrains.com/pycharm/) development environment Matplotlib Python plotting library and NumPy numerical mathematics extension.

### 3.3. Analysis of Amino Acid Sequence Alignment

WebLogo (https://weblogo.berkeley.edu) [56] was used to graphically represent the multiple amino acid sequence alignment of S1 domains. WebLogo uses the multiple sequence alignment and displays a graphical representation of sequence alignment consisting of color-coded stacks of letters representing amino acids in sequential positions. The total height of the stack indicates the sequence conservation at that particular position, while inside each stack the height of the characters indicates the relative frequency of each amino or nucleic acid at that position.

### 3.4. Statistical Analysis of the Data

The Kruskal-Wallis test [57] was used to determine the differences between the data on Bacteria, Eukaryotes, and Archaea. The result is considered insignificant at *p* > 0.05.

### 3.5. Phylogenetic Analysis of S1 Domains

The extracted protein sequences from PDB structures (Section 3.1) were used to create a phylogenetic tree using the interactive servers for multiple alignment of protein structures: POSA (http://posa.godziklab.org) [39] and SALIGN (https://modbase.compbio.ucsf.edu/salign/) [40].

### 3.6. Calculation of Radius of Gyration

The radius of gyration is defined as the root mean square distance of the collection of atoms from their common center of mass, calculated from PDB files using the module from the Pymol ScrIpt COllection library (PSICO).

## 4. Conclusions

As you know, the S1 domain is one of the “oldest” protein domains. It is presented in different combinations in different proteins, and such combinations are a direct result of the evolution of organisms. A study of the available tertiary structures of the S1 domain showed that this domain is mainly identified in eukaryotes. In this work, we studied a representative data set consisting of 24 S1 domains of different proteins to reveal their structural features necessary for the specificity of ligand binding. The main function of the bacterial S1 domain is RNA-binding through conservative amino acids on the surface of the domain. However, the functions of the bacterial ribosomal proteins S1 are not clearly defined, since the number of domains in S1 varies and is a distinctive characteristic for different phylogenetic phyla. For eukaryotes and archaea, the S1 domain almost always interacts with other domains, forming a central channel for RNA movement. Interestingly, the domains involved in the formation of pores of RNA entry, are homologous and, apparently, have an evolutionary relationship with the S1 domain. We calculated and characterized the predisposition of tertiary flexibility of S1 domains (FlexPred profiles, B-factors, and radius of gyrations) from bacterial, eukaryotic, and archaeal proteins and showed that S1 domains have similar structural features. The most flexible loop region (40–50 a.a.) in the S1 domain, apparently, can be potentially involved in the interaction of natural binding ligands. Our results show that the number of possible functions for eukaryotic proteins increases due to an increase in the number of structural domains and flexible linkers between domains, but not due to a change in the characteristics of an individual structural unit.

## Figures and Tables

**Figure 1 molecules-24-03681-f001:**
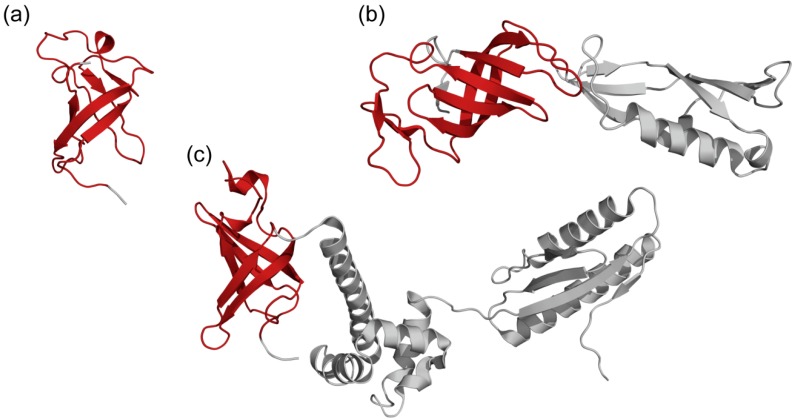
Protein structures with the S1 domain of bacterial, eukaryotic, and archaeal proteins. The S1 domain in each structure is highlighted in red. (**a**) Bacterial Polynucleotide phosphorylase (PNPase), PDB code: 1sro; (**b**) Eukaryotic RNA Polymerase II initial transcribing complex with a 2nt DNA-RNA hybrid, PDB code: 4a3g (Chain: G); (**c**) Archaeal alpha subunit of aIF2, PDB code: 1yz6.

**Figure 2 molecules-24-03681-f002:**
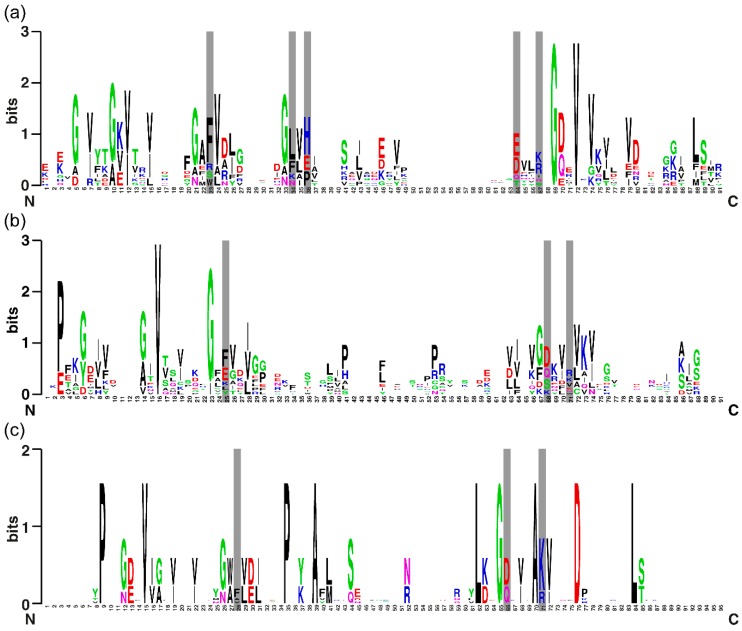
Sequence LOGO of the S1 domains for nine sequences from Bacteria (**a**), ten sequences from Eukaryotes (**b**), and five sequences from Archaea (**c**). Each LOGO is displayed in stacks of letters at each position. The total stack height is the information content of this position in bits. The height of individual letters in a stack is the probability of a letter in this position. The positions of conserved RNA-binding residues are highlighted.

**Figure 3 molecules-24-03681-f003:**
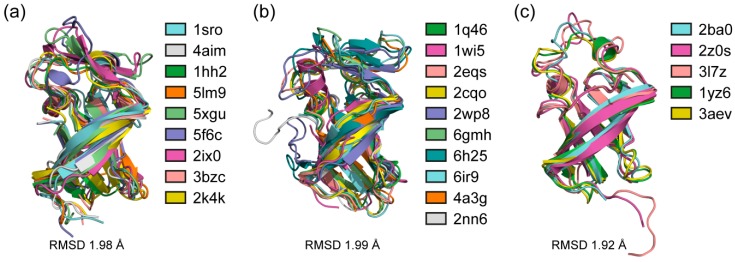
Structural alignments of S1 domains from (**a**) bacterial (nine structures); (**b**) eukaryotic (ten structures); (**c**) archaeal (five structures) proteins were made using the POSA interactive server to align multiple protein structures [39]. The average root-mean-square deviation (RMSD) for the bacterial S1 domains is 1.98 Å, for eukaryotics is 1.99 Å, for archaeal proteins is 1.92 Å.

**Figure 4 molecules-24-03681-f004:**
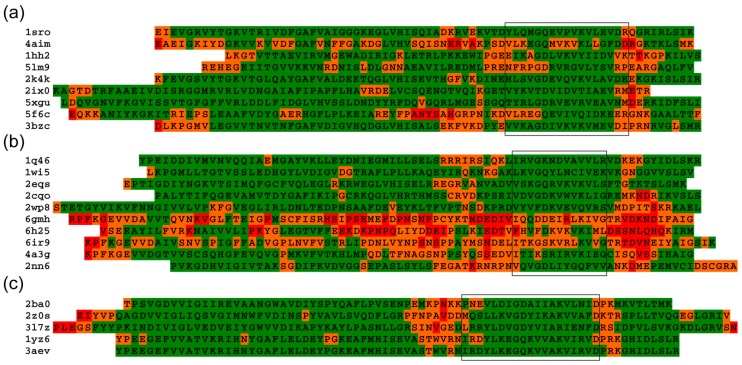
Sequences are structurally aligned relative to the framed segmented part of the S1 domains. S1 domains from (**a**) Bacteria, (**b**) Eukaryotes, and (**c**) Archaea. The colors of amino acid residues correspond to the degree of flexibility of tertiary structures according to the FlexPred software [41]: flexible (red, more than 6 Å), moderately flexible (orange, 3-6 Å), and rigid (green, less than 3Å) based on their respective values.

**Figure 5 molecules-24-03681-f005:**
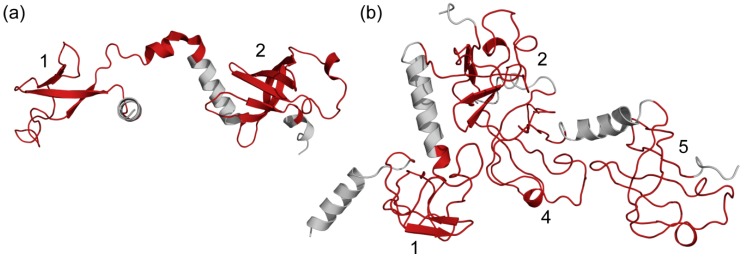
(**a**) 30S ribosomal protein S1 from *E. coli* as a part of structure of viral RNA polymerase, PDB code: 4q7j (Chain: D), (**b**) Structure of ribosomal protein S1 from *E. coli* as a part of hibernating 100S ribosome, PDB code: 6h4n (Chain: y).

**Table 1 molecules-24-03681-t001:** Bacterial proteins containing the S1 domain.

Protein Name	Source Organism	PDB Codes	Resolution	S1 Domain Function
Polynucleotide phosphorylase	*E. coli*	1sro	NMR	Promoting the initial, reversible interaction between PNPase and single-stranded RNA
	*Caulobacter vibrioides*	4aim	3.3Å (X-ray diffraction)	
Transcription termination/antitermination protein NusA	*Thermotoga maritima*	1hh2	2.1Å (X-ray diffraction)	After initial baiting of the mRNA, the S1 may pick out regulatory sequences or combinations of signals what leads to superimposition of specific binding sites on an area that nonspecifically attracts RNA
	*E. coli*	5lm9	2.14Å (X-ray diffraction)	
Ribonuclease R	*E. coli*	5xgu	1.85Å (X-ray diffraction)	S1 domain are required for binding of duplex RNA
Ribonuclease E	*E. coli*	5f6c	3Å (X-ray diffraction)	Bound RNA with the 5`-sensor domain
RNase II	*E. coli*	2ix0	2.44Å (X-ray diffraction)	RNA fragment is located in the anchor region in a deep cleft between the two CSDs and the S1 domain (loop L_45_ of S1)
Transcription accessory protein, Tex	*Pseudomonas aeruginosa*	3bzc	2.27Å (X-ray diffraction)	Tex S1 domain is required for this binding activity with a preference for ssRNA
General stress protein 13	*Bacillus subtilis*	2k4k	NMR	May can act similarly to cold shock proteins in response to cold stress

**Table 2 molecules-24-03681-t002:** Eukaryotic proteins containing the S1 domain.

Protein Name	Source Organism	PDB Codes	Resolution	S1 Domain Function
Eukaryotic translation initiation factor 2 subunit alpha	*Saccharomyces cerevisiae*	1q46	2.86Å (X-ray diffraction)	Exact function is not yet defined
Protein RRP5 homolog	*Homo sapiens*	1wi5	NMR	
ATP-dependent RNA helicase DHX8	*H. sapiens*	2eqs	NMR	
Nucleolar protein of 40 kDa	*H. sapiens*	2cqo	NMR	
Exosome complex exonuclease DIS3	*S. cerevisiae*	2wp8	3Å (X-ray diffraction)	3`end of the RNA is threaded past the S1/KH domains and through the central channel to a catalytic site
RNA polymerase II subunit G	*H. sapiens*	6gmh	3.1Å (Electron Microscopy)	Exiting RNA traverses a positively charged groove formed between the SPT6 S1 and the YqgF/RuvC domains
Exosome complex exonuclease RRP44	*H. sapiens*	6h25	3.8Å (Electron Microscopy)	RNA enters from the apical opening between the CSD lobe and the S1 domain
RNA polymerase II subunit	*Komagataella phaffii*	6ir9	3.8Å (Electron Microscopy)	Exact function is not yet defined
DNA-directed RNA polymerase II subunit RPB7	*S. cerevisiae*	4a3g	3.5Å (X-ray diffraction)	
Exosome complex component RRP40	*H. sapiens*	2nn6	3.35Å (X-ray diffraction)	

**Table 3 molecules-24-03681-t003:** Archaeal proteins containing the S1 domain.

Protein Name	Source Organism	PDB Codes	Resolution	S1 Domain Function
Exosome complex component Rrp4	*Archaeoglobus fulgidus*	2ba0	2.7Å (X-ray diffraction)	S1 domains and a subsequent neck in the RNase-PH domain ring form an RNA entry pore to the processing chamber that only allows access of unstructured RNA
	*Aeropyrum pernix*	2z0s	3.2Å (X-ray diffraction)	Exact function is not yet defined
Exosome complex component Csl4	*Saccharolobus solfataricus*	3l7z	2.41Å (X-ray diffraction)	S1 and KH domains to move away from the central channel and thus increases the diameter of the pore opening for RNA
Translation initiation factor IF-2 subunit alpha	*Pyrococcus abyssi*	1yz6	3.37Å (X-ray diffraction)	RNA binding site is formed by the concave side of the S1 barrel
	*Pyrococcus horikoshii*	3aev	2.8Å (X-ray diffraction)	

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
