# Peer review of "Investigation of the Relationship between the S1 Domain and Its Molecular Functions Derived from Studies of the Tertiary Structure"

_molecules, 2019, doi:10.3390/molecules24203681_

Round 1
Reviewer 1 Report
The paper has been substantially improved. I'm glad that authors accepted my suggestions about the analysis of percentage flexibility and the radius of gyration. Introduction is much more informative. The results are more fairly presented and realistic.
Only a few typing errors should be corrected (concatenated words)
Reviewer 2 Report
The authors have properly addressed the concerns and suggestions in my former review. The manuscript has been improved and, in my opinion, is already acceptable for publication.
This manuscript is a resubmission of an earlier submission. The following is a list of the peer review reports and author responses from that submission.
Round 1
Reviewer 1 Report
Main objection:
The paper explores the tertialy flexibility predisposition of S1 domains from bacterial, archaeal and eukaryotic proteins (24 proteins), with the goal to prove that flexibility is an important property of S1 domains. FlexPred software was used for the analysis of the C-alpha atom fluctuations, and the FoldUnfold software for structural flexibility profiles. Table 4 presents the results of the structural flexibility predictions for 15 and 11 aa segments. However, the results are NOT statistically significant. This is not in accordance with the conclusions:
For example, for 15 aa segments Kruskal-Wallis test shows that there are no differences between Bacteria (A), Archea (B) and Eucariotes (C) (P = 0.531):
Kruskal-Wallis test
H (chi2): 1,26
Hc (tie corrected): 1,267
p (same): 0,5306
Graphically, they are very similar - box and jitter (box point plot) reveals no difference
For 11 aa segments data are bimodal, and it is very hard to give definitive conclusion:
Minor objection:
The English language must be substantialy improved.
Some abbreviations are used before they were explained.

Reviewer 2 Report
This manuscript (MS) presents an interesting compared study of an important structural domain as the ancient S1 domain (involved mainly in RNA-binding) is. The work is well organized and reasonably well presented and the conclusions are supported by the structural study. However, there are some issues that should be addressed by the authors in a revised version.
In sections 2.2, 2.3, and 2.4, it would have been interesting to present an image trying to summarize the discussion presented. For example, displaying the regions functionally relevant to the enumerated proteins and showing especially the location in the structure of the mentioned conserved residues. Making use of different colors and/or graphics models (lines, sticks, spheres…), the essentials of these data could be mapped onto a “master model” of the structural S1 domain. Otherwise, those three sections are a mere catalog enumerating a list of details not easy to follow as no clear connection is apparent.
In section 2.5, Figure 2 is not particularly useful. Multiple structural alignments provide a more valuable information when a method that determines the degree of structural relationship is used. For example, methods such as DALI and SALIGN give in output dendrograms obtained from score values in the pairwise similarity matrices. These dendrograms (constructed upon purely structural input, i.e. atomic coordinates) are able to find relationships among proteins that, while sharing a same fold happen to form different clusters that very frequently reveal detailed functional and/or evolutionary similarities. This type of output in multiple structural alignments is more useful than RMSD values (Figure 2) which, for closely related proteins as it is the case with the S1 domain fold, provide no distinctive features.
The analysis of intrinsic flexibility/disorder in S1 domains is interesting as its results agree with the well-known higher frequency of protein disorder in eukaryote than in archaea and bacteria. However, the analysis appears incomplete. Firstly, given that the experimental structures of 17 out of the 24 proteins studied are crystallographic, the authors might have directly used their B-factors of alpha carbons to complement their analysis. Secondly, there are highly reliable predictors of disorder from sequence (e.g. IUPred2) whose output is a numeric score in the (0-1) range with values > 0.5 meaning disorder. This output can be easily represented graphically so that one can get a unique plot comparing the disorder profile of a number of representative sequences. Given that on the one side, the proteins studied in the MS have a relatively low number of amino acids and that on the other side, they have crystal structures these plots would allow the authors to condense their data on disorder in a graphically simple and appealing form while comparing the essential information coded in both sequence and structure. In this regard, Table S3 in the supplementary information is scarcely useful as it is a set of separate plots for the proteins studied that do not allow a direct comparison.
Other minor issues.
The date at which a search in databases is done (lines 67-68, page 2) must be specified. For example, as of July 7th, 2019, a search for “S1 domain” in the PDB gave 250 hits.
The English should be carefully revised. There are a large number of incorrect expressions (“conservative residues”, “experimentally approved”, “tertiary similarity”, “intrinsically flexibility” …) scattered over the MS that hamper understanding of some parts of the text.